# Auditing Prompt Caching in Language Model APIs

**Chenchen Gu** [1]  **Xiang Lisa Li** [1]  **Rohith Kuditipudi** [1]  **Percy Liang** [1]  **Tatsunori Hashimoto** [1]

## Abstract

Prompt caching in large language models (LLMs) results in data-dependent timing variations: cached prompts are processed faster than non-cached prompts. These timing differences introduce the risk of side-channel timing attacks. For example, if the cache is shared across users, an attacker could identify cached prompts from fast API response times to learn information about other users' prompts. Because prompt caching may cause privacy leakage, transparency around the caching policies of API providers is important. To this end, we develop and conduct statistical audits to detect prompt caching in real-world LLM API providers. We detect global cache sharing across users in seven API providers, including OpenAI, resulting in potential privacy leakage about users' prompts. Timing variations due to prompt caching can also result in leakage of information about model architecture. Namely, we find evidence that OpenAI's embedding model is a decoder-only Transformer, which was previously not publicly known.[1]

## 1. Introduction

Transformer large language models (LLMs) are computationally expensive and slow to run. To address this challenge, recent work has developed optimizations to make LLM inference and serving more efficient, such as prompt caching (Zheng et al., 2024a; Gim et al., 2024). In prompt caching, reuse of the attention key-value (KV) cache across requests results in cache hits and faster response times for prompts that share a prefix with a cached prompt.

However, prompt caching results in data-dependent timing

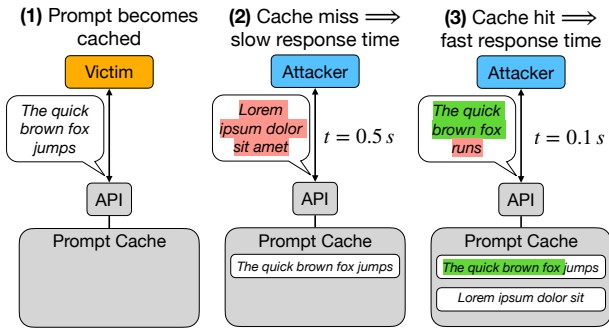

*Figure 1.* An example illustrating prompt caching. (1) A victim sends a prompt to the API, which then becomes cached. (2) An attacker sends a new prompt, resulting in a cache miss and slow response time. (3) An attacker sends a prompt that shares a prefix with the victim's prompt, resulting in a cache hit. From the fast response time, the attacker can infer that a cache hit occurred, which potentially reveals information about other users' prompts.

variations—cached prompts will be processed faster than non-cached prompts, introducing the risk of side-channel timing attacks and information leakage. In particular, an attacker could identify prompts that yield fast API response times; such prompts are likely cached. If the cache is shared across users, then a prompt being cached implies that another user recently sent that prompt. Figure 1 illustrates an example of prompt caching and potential privacy leakage. In general, timing differences between cache hits and cache misses have been widely exploited in computer security, such as in the infamous Meltdown (Lipp et al., 2018) and Spectre attacks (Kocher et al., 2019).

Because prompt caching may result in privacy leakage, it is important for users to know about the prompt caching policies of API providers. Some API providers have announced that they perform prompt caching, such as Anthropic (2024b) and OpenAI (2024b), but other API providers may be performing prompt caching without announcing it. Also, even if a provider announces prompt caching, they may not state the level of cache sharing, i.e., per-user, per-organization, or global.

Therefore, we develop and conduct an audit to determine if an API provider is caching prompts and the precise level of cache sharing. Our audit uses statistical hypothesis testing and outputs valid p-values with respect to the null hypothesis

---

[1]Stanford University. Correspondence to: Chenchen Gu <cygu@cs.stanford.edu>, Tatsunori Hashimoto <thashim@stanford.edu>.

*Proceedings of the $42^{nd}$ International Conference on Machine Learning*, Vancouver, Canada. PMLR 267, 2025. Copyright 2025 by the author(s).

[1]We release code and data at https://github.com/chenchenygu/auditing-prompt-caching.

of no caching, enabling guarantees on the false positive rate.

In our audit, we construct and sample response times from two procedures: one that attempts to produce cache hits, and one that produces cache misses. At a high level, to attempt to produce a cache hit, we send a prompt to the API to try to cache the prompt, then we send the prompt again to try to hit the cache. To produce a cache miss, we simply send a random prompt. Under the null hypothesis of no prompt caching, where only cache misses are possible, these procedures produce identical distributions of times. Accordingly, we detect caching if we find a statistically significant difference between these distributions.

We conducted audits on real-world LLM API providers in September and October 2024. We detected prompt caching in 8 out of 17 API providers. In 7 of these providers, we detected global cache sharing. On these APIs, an attacker could, in principle, detect cache hits from timing differences to infer that another user sent a prompt that shares a prefix with a given prompt.

Timing variations due to prompt caching can also result in leakage of information about a model's architecture. Cache hits between prompts that share a prefix but have different suffixes are possible only in autoregressive decoder-only Transformers, where each token attends only to previous tokens. Therefore, detecting such prompt prefix caching indicates that the model has a decoder-only architecture. Virtually all chat models are decoder-only, but embedding models can have either encoder or decoder architectures. As such, for proprietary embedding models, leakage of architecture information may represent a leakage of intellectual property. By detecting prompt prefix caching, we find evidence that OpenAI's text-embedding-3-small model has a decoder-only architecture, which was previously not publicly known.

**Responsible disclosure.** In October 2024, we disclosed our audit results with each API provider in which we detected prompt caching. We gave providers 60 days to address the vulnerabilities before publicly releasing our findings, and the actual time elapsed ended up being longer. To our knowledge, at least five providers made changes to mitigate vulnerabilities, e.g., disabling global cache sharing across organizations and updating documentation.

## 2. Preliminaries and Assumptions

First, we briefly describe prompt caching, our assumptions on how users and attackers can interact with an API, and the levels of cache sharing and privacy leakage.

### 2.1. Prompt Caching

Recent works have proposed prompt caching in Transformer (Vaswani et al., 2017) LLM serving by reusing the attention key-value (KV) cache across requests (Zheng et al., 2024a; Gim et al., 2024). In these methods, a prompt is cached by storing the prompt's attention KV cache. Then, if a subsequent prompt has a matching prefix with a cached prompt, the KV cache for the matching prefix can be retrieved from the cache. As a result, cache hits will tend to have a faster time to first token (TTFT), which is the time taken to process the prompt and generate the first response token.[2] In decoder-only Transformers, where each token attends only to previous tokens, reusing the KV cache for matching prefixes exactly preserves model behavior, even when the prompt suffixes differ. Figure 1 illustrates an example of prompt caching.

Several API providers have recently announced prompt caching features, including Anthropic (2024b), DeepSeek (2024), Fireworks (2024), and OpenAI (2024b). These providers do not state technical details of prompt caching, but these providers state that cache hits occur for (and only for) exact prefix matches between prompts. For our purposes, the precise implementation of prompt caching is largely unimportant. The properties of prompt caching that we exploit are:

1. Cache hits occur on prefix matches between prompts.

2. Cache hits tend to have a faster TTFT than cache misses (after accounting for prompt length).

To describe these properties more formally, assume that a model API takes in a prompt $x$ and has a TTFT $T(x)$. Note that $T(x)$ is a random variable due to variance from network latency, server load, etc. Assume that the API has a cache $\mathcal{C}$, which is a set of cached prompts. If $x$ has a sufficiently long matching prefix with some cached prompt $c \in \mathcal{C}$, then a cache hit occurs.

For example, let $c =$ "*The quick brown fox jumps*" and $\mathcal{C} = \{c\}$. If $x_1 =$ "*The quick brown fox runs*", then $x_1$ and $c$ have a matching prefix of "*The quick brown fox* ", so $x_1$ could result in a cache hit. On the other hand, if $x_2 =$ "*A quick brown fox jumps*", then $x_2$ and $c$ do not share a prefix, so $x_2$ results in a cache miss. Since $x_1$ and $x_2$ are similar lengths but $x_1$ is a cache hit and $x_2$ is a cache miss, we would expect that $\mathbb{E}[T(x_1)] < \mathbb{E}[T(x_2)]$.

When a prompt $x$ is sent to the API, we assume that $x$ is added to the cache $\mathcal{C}$ and that $x$ will remain in $\mathcal{C}$ for some finite period of time. The API may use multiple servers, each with their own separate caches. We do not make assump-

---

[2]In embedding models, we can view the embedding output as the first and only response "token".

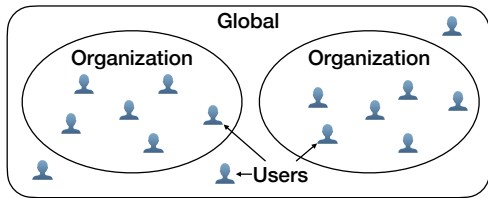

*Figure 2.* Organizations contain users, and the global level contains all users and organizations of an API.

tions about how prompts are routed to servers. A prompt may be randomly routed, or it may be intentionally routed to a server where the prompt is already cached.

## 2.2. API Assumptions

We assume that it is possible to send arbitrary prompts to the API (possibly subject to some maximum length) and measure the TTFT. The TTFT can be measured by setting the maximum tokens parameter to 1, which restricts the LLM output to only contain 1 token. Then, the overall response time is equal to the TTFT. The max tokens parameter is supported by most, if not all, real-world LLM APIs.

Either client-side or server-side timing suffices for our purposes. The client-side timing is obtained simply by measuring the time elapsed between sending the API request and receiving the API response. The server-side timing can be measured if it is contained somewhere in the API response.[3]

## 2.3. Levels of Cache Sharing and Privacy Leakage

To facilitate our discussion of prompt cache sharing and privacy leakage in APIs, we define our terminology of users and organizations. A **user** is one person that uses the API. Each user has a unique email/username and login password. An **organization** contains many users, but shares a billing system, centralized membership management, etc. Organizations can be used by companies, research groups, etc. Many, but not all, API providers support organizations, although sometimes under different terminology, such as teams or accounts. For consistency and simplicity, we refer to them all as organizations. Figure 2 shows the hierarchical structure of users and organizations.

We consider three levels of cache sharing and their corresponding potential privacy leakages.

1. **Per-user caching.** Each user has their own cache, i.e., when user $u$ sends a prompt, a cache hit can occur only with a cached prompt previously sent by user $u$. Therefore, there is no potential privacy leakage arising from per-user prompt caching.

---

2. **Per-organization caching.** Each organization has its own cache, i.e., when user $u$, who belongs to organization $o$, sends a prompt, a cache hit can occur only with a cached prompt previously sent by any user in organization $o$. There is a slight risk of privacy leakage if certain users in the organization have access to privileged information that other users should not, e.g., the CEO knowing sensitive business data. However, this risk can be mitigated, as the organization owner has full control over which users are members.

[3]We can measure server-side timing in more than half of the APIs we test, often from undocumented fields in the HTTP headers of the API response.

3. **Global caching.** The cache is shared across all users of the API, e.g., when a user sends a prompt, a cache hit can occur with any cached prompt, regardless of who sent it. This leads to the highest risk of privacy leakage, as an attacker could potentially learn information about any other user's prompts, including users in other organizations.

## 3. An Audit to Detect Prompt Caching

Next, we propose an audit to detect whether an API provider is caching prompts and determine the level of cache sharing. Our audit uses statistical hypothesis testing and outputs valid p-values with respect to the null hypothesis of no caching, allowing for guarantees on the false positive rate.

### 3.1. Audit Formulation: Statistical Hypothesis Testing

To test for a given level of cache sharing, let $u_{\text{victim}}$ and $u_{\text{attacker}}$ be two users that are the farthest away within that level. For example, to test for per-organization caching, $u_{\text{victim}}$ and $u_{\text{attacker}}$ should be different users in the same organization.

We formulate our audit as a statistical hypothesis test using the following null and alternative hypotheses:

$H_0$ : API does not cache prompts (at this level of sharing),

$H_1$ : API caches prompts (at this level of sharing).

The caching in $H_0$ does not refer only to prompt caching via the KV cache reuse described earlier. More verbosely, $H_0$ can be written as "when $u_{\text{victim}}$ sends a prompt $x$ to the API, the API does not store any information about $x$ that affects the TTFT $T(\tilde{x})$ for any future prompt $\tilde{x}$ sent by $u_{\text{attacker}}$".

To test these hypotheses, we construct procedures that attempt to produce and measure the TTFT of cache hits and cache misses. Let $\mathcal{P}$ be a distribution of prompts. To produce a cache miss, $u_{\text{attacker}}$ simply sends a random prompt $x' \sim \mathcal{P}$ to the API and measures the TTFT $T(x')$.

To attempt to produce a cache hit, first, we sample a prompt $x \sim \mathcal{P}$, and $u_{\text{victim}}$ sends $x$ to the API one or multiple times

to try to cache $x$.[4] Next, we sample $\tilde{x} \sim \mathcal{P}$ such that $\tilde{x}$ and $x$ share a prefix of a certain length. To try to produce a cache hit, $u_{\text{attacker}}$ sends $\tilde{x}$ and measures the TTFT $T(\tilde{x})$.

Let $\mathcal{D}_{\text{hit}}$ and $\mathcal{D}_{\text{miss}}$ be the distributions of TTFTs from these cache hit and cache miss procedures, respectively. Under the null hypothesis $H_0$ of no caching, $\mathcal{D}_{\text{hit}} = \mathcal{D}_{\text{miss}}$, as both procedures will produce only cache misses. In contrast, under the alternative hypothesis $H_1$ of caching, we would expect the cache hit times to tend to be faster than the cache miss times, so $\mathcal{D}_{\text{hit}} \neq \mathcal{D}_{\text{miss}}$. Now, we can reformulate our hypotheses as

$$H_0 : \mathcal{D}_{\text{hit}} = \mathcal{D}_{\text{miss}},$$
$$H_1 : \mathcal{D}_{\text{hit}} \neq \mathcal{D}_{\text{miss}}.$$

Given this reformulation, to perform our audit, we first sample TTFTs from the cache hit and cache miss procedures. Then, we run a statistical test for whether our samples came from the same distribution, e.g., the two-sample Kolmogorov-Smirnov test, producing a p-value with respect to the null hypothesis of no caching.

### 3.2. Audit Implementation Details

Next, we describe the concrete implementation details of our audit. The procedure uses the following configuration parameters: PROMPTLENGTH, PREFIXFRACTION, NUMVICTIMREQUESTS, and NUMSAMPLES. The meanings of these parameters will become clear in the descriptions below.

**Prompt distribution.** Our distribution $\mathcal{P}$ of prompts is a uniform distribution over all prompts consisting of PROMPTLENGTH English letters, lowercase and uppercase, each separated by space characters, e.g., "m x N j R". Because all commonly used byte-pair encoding (BPE) tokenizers (Gage, 1994; Sennrich et al., 2016) split on whitespace during pre-tokenization, all prompts in $\mathcal{P}$ will be exactly PROMPTLENGTH tokens long.[5]

**Cache miss.** $u_{\text{victim}}$ sends a random prompt $x \sim \mathcal{P}$ to the API and measures the TTFT $T(x)$. Since the prompt consists of random letters, there is a negligible probability that a noticeable prefix has already been cached: the probability that two prompts sampled from $\mathcal{P}$ share a prefix of 15 tokens or longer is less than $10^{-25}$. Therefore, this procedure accurately measures a distribution of cache miss times.

---

[4]Multiple requests may be necessary in some scenarios, e.g., if API requests are randomly routed to one of several servers, and each server has a separate cache.

[5]Many APIs add a small number of tokens to the user prompt due to the default system prompt, special tokens for prompt and role formatting, etc. However, these additional tokens are unimportant for our procedure, as the number of additional tokens is small and remains constant across prompts to a given model API.

**Cache hit.** First, we sample a random prompt $x \sim \mathcal{P}$. Then, $u_{\text{victim}}$ sends $x$ to the API NUMVICTIMREQUESTS times consecutively to try to cache $x$. Then, we sample $\tilde{x} \sim \mathcal{P}$ such that $\tilde{x}$ and $x$ have a shared prefix of exactly PREFIXFRACTION $\times$ PROMPTLENGTH tokens. To attempt to produce a cache hit, $u_{\text{attacker}}$ sends $\tilde{x}$ to the API and measures the TTFT $T(\tilde{x})$. When PREFIXFRACTION $= 1$, we test for prompt caching when $\tilde{x} = x$, i.e., exact prompt matches. When PREFIXFRACTION $< 1$, we test for prompt caching when $\tilde{x}$ and $x$ have the same prefix but different suffixes, e.g., "a b c d" and "a b c x".

**Statistical testing.** Putting these pieces together, to perform the audit, we collect NUMSAMPLES timings each from the cache hit and cache miss procedures. We randomize the order in which we collect the timing samples. Then, we test for a statistically significant difference between the distributions of times from the two procedures. We use the SciPy implementation (Virtanen et al., 2020) of the two-sample Kolmogorov-Smirnov (KS) test (Hodges Jr, 1958), which is a nonparametric test for equality of distributions. The test statistic is the maximum difference between the empirical cumulative distribution functions at any point. More specifically, since we expect cache hits to be faster under the alternative, we perform a one-sided test, so the test statistic is the maximum difference in the direction of cache hits being faster. The KS test outputs a p-value, which we can use to reject or not reject the null hypothesis of no prompt caching at a given significance level $\alpha$.

## 4. Auditing Real-World APIs

Next, we audit real-world LLM APIs to identify APIs that cache prompts and determine the level of cache sharing, i.e., per-user, per-organization, or global. Cache sharing results in potential privacy leakage, as an attacker could, in principle, identify cached prompts using timing data to learn information about other users' prompts.

### 4.1. Audit Setup and Configuration

**API providers and models.** We audit 17 API providers: Anthropic, Amazon Bedrock, Microsoft Azure OpenAI, Cohere, Deep Infra, DeepSeek, Fireworks AI, Google, Groq, Hyperbolic, Lepton AI, Mistral, OctoAI, OpenAI, Perplexity, Replicate, and Together AI. The model APIs that we audit for each provider are included in Tables 1 and 2. For API providers that primarily serve open-weight models, we audit their Llama 3 or 3.1 8B Instruct API (Dubey et al., 2024). For providers that serve proprietary models, we audit the cheapest chat model in their most recent family of models. In addition, we audit APIs for proprietary embedding models, where available. We do not audit APIs for open-weight embedding models because we did not find any

*Table 1.* Audit results for APIs where we detected prompt caching. ✓ denotes caching was detected, ✗ denotes caching was not detected, and "—" denotes that cache sharing within an organization was not tested, either because the API did not support organizations or because we did not have access to the organizations feature. We report the average precision for classifying times from the cache hit procedure, using the highest level of cache sharing detected in each API. We report the average precision for client-side timing and server-side timing separately, with "—" denoting that the given timing method is unavailable for that API.

| | | Same prompt | Same prefix, different suffixes | | | Avg. precision | |
|---|---|---|---|---|---|---|---|
| Provider | Model | Per-user | Per-user | Per-org. | Global | Client | Server |
| Azure | text-embedding-3-small | ✓ | ✓ | — | ✓ | 0.80 | — |
| Deep Infra | Llama 3.1 8B Instruct | ✓ | ✓ | — | ✓ | 0.84 | — |
| Fireworks | Llama 3.1 8B Instruct | ✓ | ✓ | ✓ | ✓ | 0.77 | 0.79 |
| Lepton | Llama 3.1 8B Instruct | ✓ | ✓ | — | ✓ | 0.71 | 0.70 |
| OpenAI | text-embedding-3-small | ✓ | ✓ | ✓ | ✓ | 0.78 | 0.79 |
| Perplexity | Llama 3.1 8B Instruct | ✓ | ✓ | — | ✓ | 0.90 | — |
| Replicate | Llama 3 8B Instruct | ✓ | ✓ | — | ✓ | — | 1.00 |
| Anthropic | Claude 3 Haiku | ✓ | ✓ | ✓ | ✗ | 0.84 | — |
| OpenAI | GPT-4o mini | ✓ | ✓ | ✓ | ✗ | 0.79 | 0.86 |

*Table 2.* Audit results for APIs where we did not detect prompt caching. ✗ denotes that caching was not detected.

| | | Same prompt |
|---|---|---|
| Provider | Model | Per-user |
| Amazon | Claude 3 Haiku | ✗ |
| Azure | GPT-4o mini | ✗ |
| Cohere | Command R | ✗ |
| Cohere | embed-english-v3.0 | ✗ |
| DeepSeek | DeepSeek Chat | ✗ |
| Google | Gemini 1.5 Flash | ✗ |
| Google | text-embedding-004 | ✗ |
| Groq | Llama 3 8B Instruct | ✗ |
| Hyperbolic | Llama 3.1 8B Instruct | ✗ |
| Mistral | Mistral Nemo | ✗ |
| Mistral | Mistral Embed | ✗ |
| OctoAI | Llama 3.1 8B Instruct | ✗ |
| Together | Llama 3.1 8B Instruct | ✗ |

APIs that served open-weight decoder-only Transformer embedding models. Prefix caching is possible in decoder-only Transformers but not encoder-only Transformers, where each token attends to all other tokens in the prompt.

**Configuration and procedure.** For our audits, we use PROMPTLENGTH = 5000 and NUMSAMPLES = 250. We run four levels of audits of increasing cache sharing and privacy leakage. At each level, we only continue to audit APIs if we detect caching during the previous level. We use a significance level of $\alpha = 10^{-8}$.

To narrow down our list of providers, the first level tests for the simplest level of prompt caching:

1. **Same prompt, per-user caching.** We test for prompt caching on exact prompt matches ($\tilde{x} = x$)

by setting PREFIXFRACTION = 1. We set $u_{\text{victim}}$ and $u_{\text{attacker}}$ to be the same user, and we set NUMVICTIMREQUESTS = 25.

In the remaining three levels, we test for prompt caching when $\tilde{x}$ and $x$ have the **same prefix but different suffixes** by setting PREFIXFRACTION = 0.95. We test for increasing levels of cache sharing by appropriately setting the victim and attacker users:

2. **Per-user caching.** $u_{\text{victim}}$ and $u_{\text{attacker}}$ are the same user, as in the first level.

3. **Per-organization caching.** $u_{\text{victim}}$ and $u_{\text{attacker}}$ are different users within the same organization. For APIs without organizations, we skip this level.

4. **Global caching.** $u_{\text{victim}}$ and $u_{\text{attacker}}$ are different users in different organizations. For APIs without organizations, $u_{\text{victim}}$ and $u_{\text{attacker}}$ are simply different users.

In levels 2–4, to determine how many victim requests are needed to detect caching, we run tests using NUMVICTIMREQUESTS $\in \{1, 5, 25\}$ in increasing order, stopping after the first significant p-value. To account for multiple testing, we perform a Bonferroni correction by dividing the significance threshold for each test by three.

In all levels, if only one timing method is available in an API (client-side or server-side timing), then we use that timing method. If both are available, we run tests using both timing methods and perform another Bonferroni correction, dividing by two this time.

**Cost per test.** When NUMVICTIMREQUESTS = 25, one test uses roughly 34 million prompt tokens. The number of

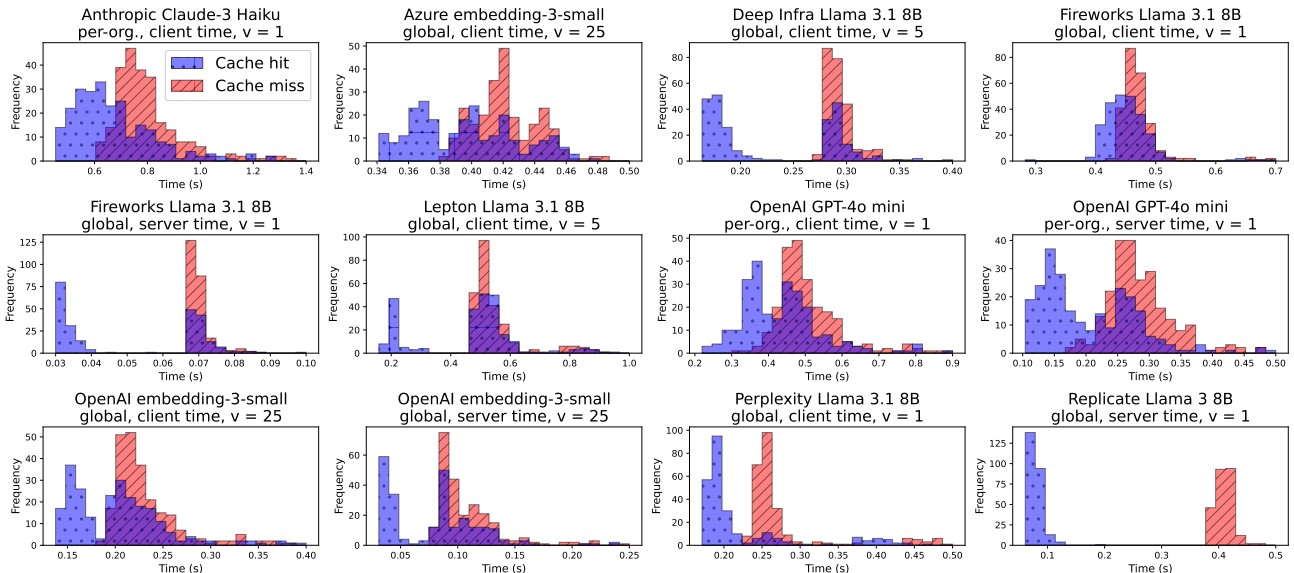

*Figure 3.* Histograms of response times from the cache hit and cache miss procedures in APIs where we detected caching. The distributions of times are clearly distinguishable, with cache hits tending to be faster. Each histogram title states the API provider, model, level of cache sharing (per-org. or global), timing source (client-side or server-side timing), and the NUMVICTIMREQUESTS used, denoted V.

response tokens used is much smaller because we set the maximum response tokens parameter to 1. For the chat APIs we audit, the prices per million prompt tokens are 0.05–0.25 USD, resulting in a cost per test of 1.69–8.44 USD. The tests are cheaper when NUMVICTIMREQUESTS is smaller.

### 4.2. Audit Results

We conducted our audits in September and early October 2024 using clients located in California. Table 1 shows audit results for APIs in which we detected prompt caching, and Table 2 shows APIs in which we did not detect prompt caching. We detected prompt caching in 8 out of 17 API providers, and we detected global cache sharing in 7 providers. This means that an attacker can potentially learn information about other users' prompts by identifying cached prompts from timing data. To assess an attacker's ability to distinguish between cache hits and cache misses, Figure 4 contains selected precision-recall curves for classifying times from the cache hit procedure.[6] The curves show that cache hits can be detected with near perfect precision up to moderate recall scores. Figure 6 in the appendix shows precision-recall curves for the highest level of cache sharing we detected in each API. To numerically summarize these curves, we compute the average precision (Zhu, 2004), which is equal to the area under the precision-recall curve (the precision is averaged over the interval of recall scores from 0 to 1). Table 1 shows that the average precisions mostly lie around a moderately high value of 0.8.

Figure 3 displays histograms of times from the cache hit and cache miss procedures. The distributions of times are clearly distinguishable, with cache hits tending to be faster. Each histogram title states the minimum NUMVICTIMRE-QUESTS (denoted V in the titles) that resulted in a significant p-value. In most of the APIs where we detected caching, only NUMVICTIMREQUESTS = 1 was needed to detect caching. Only the OpenAI and Azure text-embedding-3-small APIs required NUMVICTIMREQUESTS = 25 to achieve a significant p-value. This may suggest that these APIs have multiple servers with separate caches and that requests are randomly routed to a server, so multiple victim requests are needed to cache the prompt in enough servers for the attacker's prompt to have a sufficient probability of producing a cache hit. In Appendix B, we report all the p-values from our audits. In many APIs, the p-values are many orders of magnitude smaller than our significance level of $\alpha = 10^{-8}$. In all APIs where we detected caching, all available timing methods resulted in significant p-values.

In the Anthropic Claude 3 Haiku and OpenAI GPT-4o mini APIs, we detected per-organization cache sharing, but not global cache sharing. This exact level of cache sharing is stated in their prompt caching documentations, confirming the efficacy of our audit procedure. Since OpenAI (2024a) and Anthropic (2024a) document per-organization cache sharing, we do not consider it a security vulnerability. Global cache sharing in the OpenAI text-embedding-3-small API was a potential vulnerability, but has been patched after our responsible disclosure prior to the release of this paper.

Although DeepSeek (2024) has a prompt caching feature and returns the number of cache hit tokens in API responses,

---

[6]The cache hit procedure attempts to produce cache hits but cannot guarantee cache hits (e.g., due to server routing), so some times in the cache hit distribution may actually be cache misses.

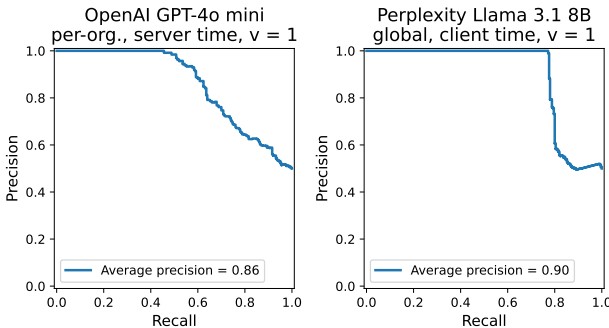

Figure 4. Selected precision-recall curves for distinguishing between times from the cache hit and cache miss procedures. Cache hits are the positive class. The curves show that cache hits can be detected with near perfect precision up to moderate recall scores. Figure 6 in the appendix contains curves for other APIs.

which we used to confirm that we produced cache hits, we were unable to detect caching from response times. There was no statistically significant difference between the distributions of cache hit and cache miss times, even in two-sided tests. DeepSeek states that the cache is isolated per-user, and we empirically verified that this is the case based on the number of cache hit tokens returned in the API responses.

### 4.3. Ablations

We run ablations to determine the effects of PROMPTLENGTH, PREFIXFRACTION, and model size on the average precision, shown in Figure 5. We use the APIs in which we detected global caching with NUMVICTIMREQUESTS = 1, i.e., the Llama 3 or 3.1 8B Instruct APIs of Fireworks, Perplexity, and Replicate.

**Smaller PROMPTLENGTH decreases average precision.** In Figures 5a and 5b, we vary the PROMPTLENGTH in the same prompt (PREFIXFRACTION = 1) and same prefix but different suffixes (PREFIXFRACTION = 0.95) settings, respectively. When the PROMPTLENGTH is moderately high ($\gtrsim 1000$), the average precision is relatively high and stable. However, as the PROMPTLENGTH approaches zero, the average precision decreases to random chance.

**Decreasing PREFIXFRACTION decreases average precision.** In Figure 5c, we vary the PREFIXFRACTION while setting PROMPTLENGTH = 1000. As the length of the matching prefix decreases, the average precision decreases to random chance.

**No clear relationship between model size and average precision.** In Figure 5d, we vary the model size on the Fireworks API, which supports all models in the Llama 3.1 and 3.2 families. We detected caching in all model sizes, with no clear relationship between model size and average precision.

**Relationship to p-values.** Figure 7 in Appendix C shows the effects of the ablations on the audit p-values. We observe similar patterns as above, with decreases in average precision corresponding to increases in p-values.

### 4.4. Difficulty of Prompt Extraction Attacks

Our results show that given a specific prompt $x$, an attacker could potentially detect cache hits to learn whether another user sent a prompt that shares a prefix with $x$. A natural question is whether an attacker could extract other users' prompts token-by-token. One idea is to use breadth-first search: given a partial candidate prompt, such an attack would try possible continuation tokens and determine which continuation token is cached. The cached token is appended to the candidate prompt and the process repeats.

However, we were unable to execute practical prompt extraction attacks. A successful attack requires extremely accurate detection of cached tokens, as there are many possible continuation tokens at each step. Just one incorrect token causes complete failure due to the exact prefix match required for a cache hit. In preliminary experiments, we were unable to reliably detect the presence of one additional cached token. It is also difficult to make repeated measurements to boost accuracy. To detect whether a prompt is cached, the attacker must send the prompt to the API. Then, future measurements may produce a cache hit not because another user sent the prompt, but because the attacker sent it.

We do not claim that prompt extraction attacks are necessarily impossible. Such attacks face difficulties, but future work may yet develop successful, practical attacks. In addition, in more restricted sets of target prompts, e.g., known prompt templates with places for users to enter private personal information, it may be easier to overcome these difficulties.

## 5. Leakage of Architecture Information

In addition to privacy implications, the detection of prompt caching can also reveal information about a model's architecture. This is because the conditions for cache hits to occur depend on model architecture.

In decoder-only Transformer models, reuse of the attention KV cache enables cache hits between prompts with matching prefixes, even if the suffixes differ, since each token attends only to previous tokens. This prefix caching is not possible in encoder-only or encoder-decoder Transformer models, where each token in the prompt attends to all other tokens in the prompt. Therefore, detecting such prompt prefix caching indicates that a model cannot have a bidirectional encoder architecture. Virtually all chat models are decoder-only, but embedding models can have either encoder or decoder architectures, as seen in the Massive Text Embedding Benchmark (MTEB) leaderboard (Muennighoff

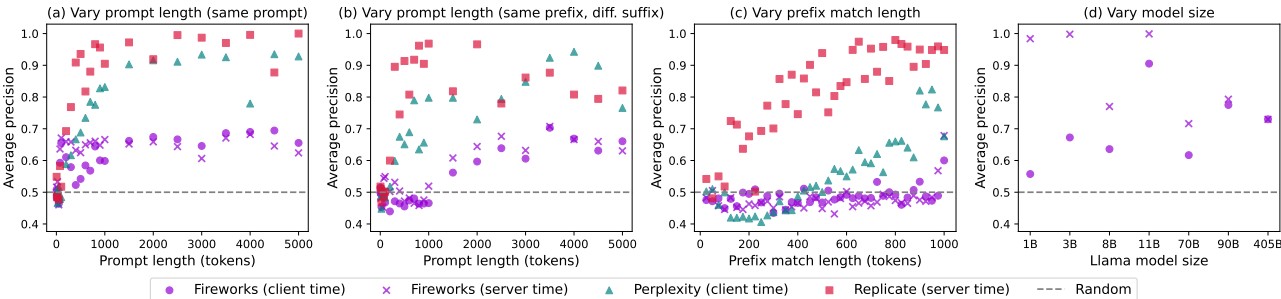

*Figure 5.* Ablations on the effects of PROMPTLENGTH, PREFIXFRACTION, and model size on the average precision. In (a)–(c), as the prompt length or prefix match length decreases, the average precision decreases to random chance. In (d), we detect caching across all model sizes, with no clear relationship between model size and average precision.

et al., 2023). As such, for proprietary embedding models, leakage of architecture information may represent a leakage of intellectual property.

In our audits (Table 1), we detected prompt caching in OpenAI's text-embedding-3-small API when prompts had the same prefix but different suffixes. We confirm that when the prompt suffix is changed, the returned embedding also changes, indicating that the caching mechanism does not simply return cached embedding outputs from similar prompts. Assuming that text-embedding-3-small is Transformer-based, this indicates that text-embedding-3-small is a decoder-only Transformer. This is new information, as OpenAI has not released any information about the architecture of their embedding models.

**Floating-point precision of the cache.** When we send the exact same prompt multiple times, when the response time is noticeably faster, indicating a cache hit, the returned embedding differs slightly from the "normal" embedding in most of the responses with normal response times, which indicate cache misses. This behavior is consistent across different random prompts. These differences are small, on the order of $10^{-4}$ to $10^{-5}$ in each coordinate. We hypothesize that these differences may arise if the reused KV cache is stored in a lower floating-point precision, resulting in slight discrepancies when the attention KV is computed from scratch in cache misses versus when it is retrieved from the cache in cache hits. Interestingly, in some responses, especially those that are noticeably slower, the embedding differs from both the "normal" and "cache hit" embeddings. This may be caused by some responses being processed by different GPU models, as floating point computations can differ slightly across different GPUs. Appendix D contains examples of response times and embeddings showing this phenomenon.

## 6. Mitigations

**Per-user caching prevents privacy leakage.** To completely prevent any privacy leakage from prompt caching, only per-user caching should be allowed. In per-user

caching, an attacker will not be able to produce cache hits on prompts sent by other users. Since it is unlikely that different users will send prompts with long matching prefixes, per-user caching should retain many of the performance benefits from global cache sharing.

**Disclosure of cache sharing.** We believe that providers should disclose their caching policies, particularly the level of cache sharing. It is important that users know how their data is handled and who could potentially learn information about their data. This way, users can make informed decisions about how they use an LLM API. For example, if a company knows that an API uses per-organization cache sharing, the company can decide to create separate organizations for different groups of employees to prevent unauthorized information access.

**Disabling caching prevents any information leakage.** For information leakage that only requires per-user caching, such as leakage of architecture information, the strongest mitigation is to disable prompt caching. Since per-user caching does not result in privacy leakage, but may result in leakage of the API provider's intellectual property, it is up to the provider to determine their level of risk tolerance. Another potential mitigation is to intentionally delay the response time for cache hits so that they look like cache misses. This eliminates the benefits of prompt caching for users, but API providers could still benefit, as cached prompts require less GPU processing time.

## 7. Related Work

**Prompt caching.** Many recent works have developed optimizations for inference and serving of Transformer language models. Various methods involve reuse of the attention KV cache, improving latency and throughput for shared prompt prefixes (Kwon et al., 2023; Zheng et al., 2024a; Gim et al., 2024; Ye et al., 2024a;b; Qin et al., 2024; Juravsky et al., 2024). Recall that we do not assume any particular implementation of prompt caching in our attacks. Indeed, we do not know technical details about the caching mechanisms

used by the APIs we audited. Other caching methods do not preserve exact model behavior, such as retrieving cached responses for semantically similar prompts (Bang, 2023) or reusing the KV cache even when the prefixes do not exactly match (Gim et al., 2024; Yao et al., 2025; Hu et al., 2024). We do not study such methods, but they are also likely susceptible to similar cache timing attacks, and our audit can easily be adapted to detect other types of caching.

**Cache timing attacks.** In computer security, many side-channel timing attacks have extracted information by using timing differences to distinguish between cache hits and cache misses, e.g., in the CPU cache or web cache. For example, cache timing attacks have been used to extract AES keys (Bernstein, 2005; Osvik et al., 2006; Bonneau & Mironov, 2006; Tromer et al., 2010; Gullasch et al., 2011; Yarom et al., 2017), a user's private web information (Felten & Schneider, 2000; Bortz & Boneh, 2007; Van Goethem et al., 2015), and sensitive data from other processes on a machine (Percival, 2005; Yarom & Falkner, 2014; Liu et al., 2015), as in the well-known Meltdown (Lipp et al., 2018) and Spectre attacks (Kocher et al., 2019).

**Attacks on language model APIs.** Several recent works have attacked language model APIs. Carlini et al. (2024) and Finlayson et al. (2024) show that logits and logprobs leak information from an LLM API, including the model's hidden dimension size and final layer weights. Weiss et al. (2024) partially extract encrypted and streamed LLM responses by inferring and analyzing token lengths from packet sizes. Carlini & Nasr (2024) and Wei et al. (2024) exploit speculative decoding (Leviathan et al., 2023; Chen et al., 2023) and similar methods to extract LLM responses with higher success by measuring delays between packets.

Most related to our work are Song et al. (2024) and Zheng et al. (2024b), which also study timing attacks and privacy leakages arising from prompt caching, including both KV cache reuse and semantic caching, primarily in simulated, controlled environments. Our work differs in developing an audit that is practical and provides statistical guarantees, using these audits to precisely identify different levels of cache sharing, and extracting information about model architecture. Song et al. (2024) demonstrate prompt extraction attacks in a simulated setting, but the attack is run locally without network latency, uses knowledge of the distribution of prompts, requires explicit clearing of the cache to make repeated measurements, and makes an average of over 200 measurements for each extracted token. Due to these limitations, we believe that these simulated attacks are currently unlikely to be real-world privacy threats.

## 8. Conclusion

As LLMs and other machine learning systems become more widely deployed and used in the real world, it is increasingly important to consider security and privacy aspects of these systems. To this end, in this paper, we find that prompt caching in LLM APIs can leak private and proprietary information through timing differences. We develop and conduct rigorous statistical audits on real-world APIs, finding that multiple APIs were performing global cache sharing. We hope that future work will continue to evaluate and audit the security and privacy of machine learning systems, ensuring their robustness and trustworthiness.

## Impact Statement

**Responsible disclosure.** As discussed earlier, to mitigate real-world harms arising from our research, we performed responsible disclosure. In October 2024, we disclosed our audit results with each API provider in which we detected prompt caching. We gave providers 60 days to address the vulnerabilities before publicly releasing our findings, and the actual time elapsed ended up being longer. To our knowledge, at least five providers made changes to mitigate vulnerabilities, e.g., disabling global cache sharing across organizations and updating documentation.

**Broader impact.** We believe that our audits for detecting prompt caching and the level of cache sharing in LLM APIs can improve transparency and trust. By increasing transparency around caching policies and how user data is handled, users can make better informed decisions about how they use an LLM API and have the appropriate level of trust that their data will be secure and private. More broadly, we believe that audits are a promising method to ensure that machine learning systems are safe, secure, and trustworthy, especially as these systems become more widely deployed and have larger societal impact.

## Acknowledgments

CG was supported by the Stanford CURIS program. XL was supported by a Two Sigma PhD Fellowship. PL was supported by NSF Award Grant no. 1805310 and an Open Philanthropy Project Award. TH was supported by gifts from Open Philanthropy, Amazon, Google, Meta, and a grant under the NSF CAREER IIS-2338866.

## Conflicts of Interest

PL is a co-founder of Together AI. However, this work was done in his Stanford capacity. The methods, providers audited, and results were not influenced by or shared with Together prior to the public release of this paper. All API

providers were audited using the same procedure, including Together. When this work was conducted, none of the other authors had conflicts of interest with the providers audited in this paper.

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

## A. Precision-Recall Curves

Figure 6 shows precision-recall curves for distinguishing between cache hit and cache miss times in APIs where we detected caching in our audits (Table 1).

## B. P-values from Audits

We report all the p-values from our audits on APIs. Table 3 contains p-values from level 1 of our audits: same prompt, per-user caching. Table 4 contains p-values from level 2 of our audits: prompts with the same prefix but different suffixes, per-user caching. Table 5 contains p-values from level 3 of our audits: prompts with the same prefix but different suffixes, per-organization caching. Table 6 contains p-values from level 4 of our audits: prompts with the same prefix but different suffixes, global caching.

## C. Ablation Effects on Audit p-values

Figure 7 shows the effects of the ablations in §4.3 on the audit p-values. Each test is run using NUMSAMPLES $= 250$. We observe similar patterns as in §4.3, with decreases in average precision corresponding to increases in p-values. As the prompt length or prefix match length decreases, the p-values grow larger. We detect caching across all model sizes, with no clear relationship between model size and p-values.

## D. Embeddings and Response Times from the OpenAI text-embedding-3-small API

Tables 7, 8, and 9 contain real examples of response times and embeddings illustrating the phenomenon discussed in §5. Each table contains the server-side response times and first five embedding coordinates when sending the same prompt 25 consecutive times to the OpenAI text-embedding-3-small API from the same user. More specifically, each table shows the victim requests for one prompt in the main audits, which used PROMPTLENGTH $= 5000$.

The tables show that there is a "normal" embedding (shown in blue) that is returned in most of the responses with normal response times, which indicate cache misses. When the response time is noticeably faster (shown in green), indicating a cache hit, the embedding differs slightly from the "normal" embedding. Most, but not always all, fast responses have the same embedding. In some responses, especially those that are noticeably slower, the embedding (shown in red) differs from both the "normal" and "cache hit" embeddings. There are several different "alternate" embeddings among these responses.

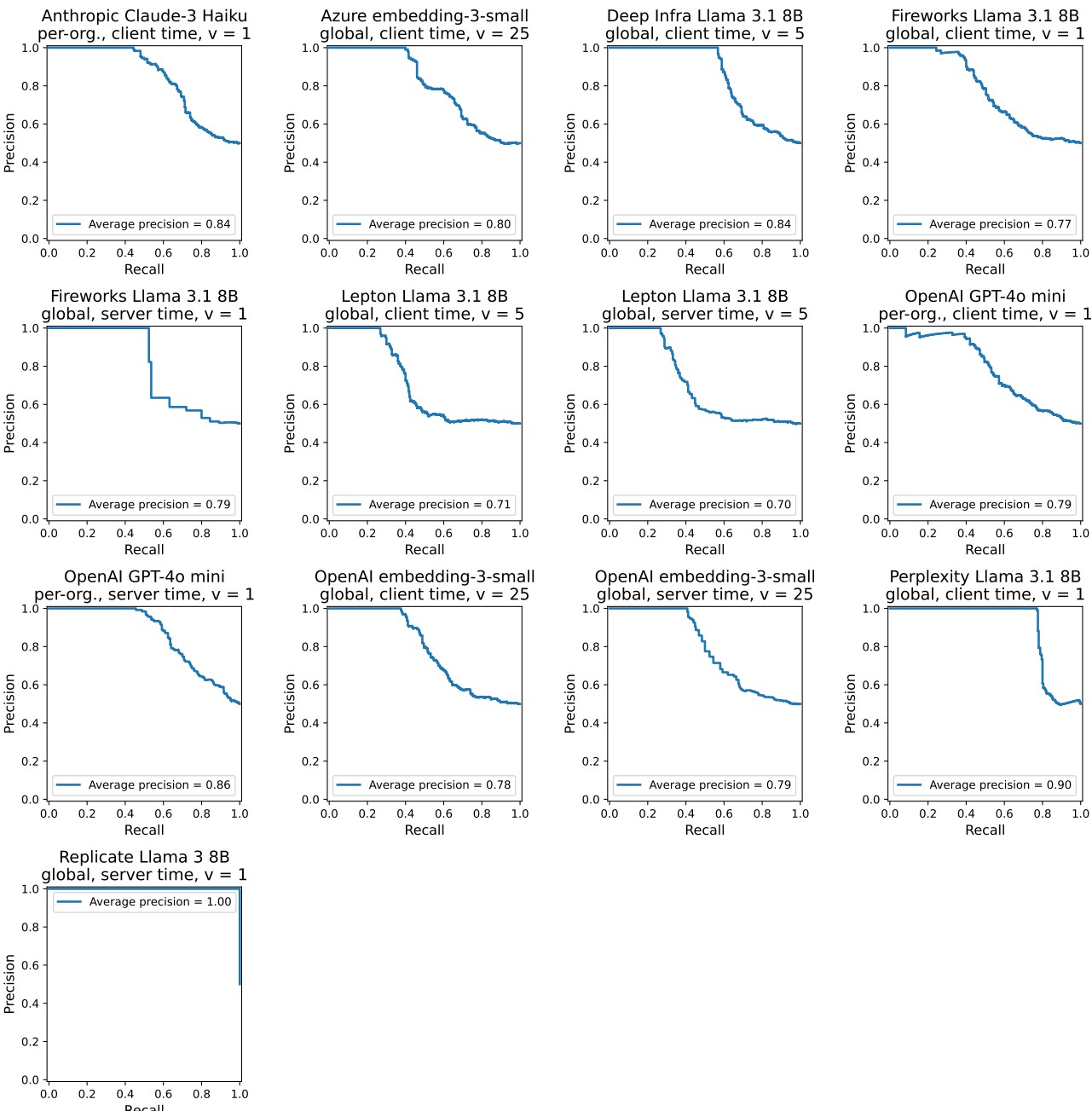

*Figure 6.* Precision-recall curves for distinguishing between times produced by the cache hit and cache miss procedures in APIs where we detected caching in our audits (Table 1). Cache hits are the positive class, and cache misses are the negative class. The curves show that cache hits can be detected with near perfect precision up to moderate recall scores. Note that our cache hit procedure attempts to produce cache hits but cannot guarantee cache hits (e.g., due to server routing), so some times in the cache hit distribution may actually be cache misses, which would hurt recall scores.

*Table 3.* P-values from level 1 of our audits: same prompt, per-user caching. Each column shows one combination of NUMVICTIMRE-QUESTS and timing source (client-side or server-side timing). Green indicates a significant p-value, after performing the appropriate Bonferroni corrections. Red indicates a p-value that is not significant. "—" indicates that the given timing source was not available for the API. APIs are grouped by whether caching was detected in this level and sorted alphabetically within the groups.

| | | NUMVICTIMREQUESTS 25 | |
| | | Client | Server |
| Provider | Model | | |
|---|---|---|---|
| Anthropic | Claude 3 Haiku | $7.8 \times 10^{-21}$ | — |
| Azure | text-embedding-3-small | $1.7 \times 10^{-42}$ | — |
| Deep Infra | Llama 3.1 8B Instruct | $9.5 \times 10^{-116}$ | — |
| Fireworks | Llama 3.1 8B Instruct | $2.0 \times 10^{-80}$ | $4.7 \times 10^{-109}$ |
| Lepton | Llama 3.1 8B Instruct | $2.2 \times 10^{-138}$ | $2.2 \times 10^{-138}$ |
| OpenAI | GPT-4o mini | $2.4 \times 10^{-66}$ | $2.9 \times 10^{-105}$ |
| OpenAI | text-embedding-3-small | $7.6 \times 10^{-9}$ | $2.3 \times 10^{-10}$ |
| Perplexity | Llama 3.1 8B Instruct | $1.9 \times 10^{-90}$ | — |
| Replicate | Llama 3 8B Instruct | — | $2.2 \times 10^{-140}$ |
| Amazon | Claude 3 Haiku | 0.27 | 0.51 |
| Azure | GPT-4o mini | 0.95 | — |
| Cohere | Command R | 0.62 | 0.72 |
| Cohere | embed-english-v3.0 | 0.41 | 0.56 |
| DeepSeek | DeepSeek Chat | 0.75 | — |
| Google | Gemini 1.5 Flash | 0.17 | 0.20 |
| Google | text-embedding-004 | 0.20 | 0.24 |
| Groq | Llama 3 8B Instruct | 0.41 | 0.51 |
| Hyperbolic | Llama 3.1 8B Instruct | 0.72 | — |
| Mistral | Mistral Nemo | 0.56 | 0.96 |
| Mistral | Mistral Embed | 0.67 | 0.91 |
| OctoAI | Llama 3.1 8B Instruct | 0.32 | 0.27 |
| Together | Llama 3.1 8B Instruct | 0.51 | 0.96 |

*Table 4.* P-values from level 2 of our audits: prompts with the same prefix but different suffixes, per-user caching. Each column shows one combination of NUMVICTIMREQUESTS and timing source (client-side or server-side timing). Green indicates a significant p-value, after performing the appropriate Bonferroni corrections. Red indicates a p-value that is not significant. "—" indicates that the given timing source was not available for the API. A blank cell indicates that the given value of NUMVICTIMREQUESTS was not tested because caching was detected in the API using a smaller value of NUMVICTIMREQUESTS. Caching was detected in all APIs audited in this level. APIs are sorted alphabetically.

| | | NUMVICTIMREQUESTS | | | | | |
| | | 1 | | 5 | | 25 | |
| Provider | Model | Client | Server | Client | Server | Client | Server |
|---|---|---|---|---|---|---|---|
| Anthropic | Claude 3 Haiku | $9.6 \times 10^{-37}$ | — | | | | |
| Azure | text-embedding-3-small | 0.20 | — | $6.0 \times 10^{-4}$ | — | $6.9 \times 10^{-42}$ | — |
| Deep Infra | Llama 3.1 8B Instruct | 0.03 | — | $5.0 \times 10^{-22}$ | — | | |
| Fireworks | Llama 3.1 8B Instruct | $4.3 \times 10^{-15}$ | $5.0 \times 10^{-33}$ | | | | |
| Lepton | Llama 3.1 8B Instruct | 1.00 | 0.96 | $7.7 \times 10^{-10}$ | $7.7 \times 10^{-10}$ | | |
| OpenAI | GPT-4o mini | $9.5 \times 10^{-27}$ | $1.5 \times 10^{-39}$ | | | | |
| OpenAI | text-embedding-3-small | 0.03 | 0.03 | 0.10 | 0.17 | $2.6 \times 10^{-12}$ | $4.3 \times 10^{-15}$ |
| Perplexity | Llama 3.1 8B Instruct | $5.4 \times 10^{-68}$ | — | | | | |
| Replicate | Llama 3 8B Instruct | — | $8.6 \times 10^{-150}$ | | | | |

*Table 5.* P-values from level 3 of our audits: prompts with the same prefix but different suffixes, per-organization caching. Each column shows one combination of NUMVICTIMREQUESTS and timing source (client-side or server-side timing). Green indicates a significant p-value, after performing the appropriate Bonferroni corrections. Red indicates a p-value that is not significant. "—" indicates that the given timing source was not available for the API. A blank cell indicates that the given value of NUMVICTIMREQUESTS was not tested because caching was detected in the API using a smaller value of NUMVICTIMREQUESTS. Caching was detected in all APIs audited in this level. APIs are sorted alphabetically.

| | | NUMVICTIMREQUESTS | | | | | |
| | | 1 | | 5 | | 25 | |
| Provider | Model | Client | Server | Client | Server | Client | Server |
|---|---|---|---|---|---|---|---|
| Anthropic | Claude 3 Haiku | $1.7 \times 10^{-31}$ | — | | | | |
| Fireworks | Llama 3.1 8B Instruct | $1.3 \times 10^{-21}$ | $5.2 \times 10^{-32}$ | | | | |
| OpenAI | GPT-4o mini | $1.1 \times 10^{-19}$ | $4.6 \times 10^{-34}$ | | | | |
| OpenAI | text-embedding-3-small | 0.27 | 0.14 | 0.27 | 0.27 | $8.2 \times 10^{-14}$ | $8.2 \times 10^{-14}$ |

*Table 6.* P-values from level 4 of our audits: prompts with the same prefix but different suffixes, global cache sharing. Each column shows one combination of NUMVICTIMREQUESTS and timing source (client-side or server-side timing). Green indicates a significant p-value, after performing the appropriate Bonferroni corrections. Red indicates a p-value that is not significant. "—" indicates that the given timing source was not available for the API. A blank cell indicates that the given value of NUMVICTIMREQUESTS was not tested because caching was detected in the API using a smaller value of NUMVICTIMREQUESTS. APIs are grouped by whether caching was detected in this level and sorted alphabetically within the groups.

| | | NUMVICTIMREQUESTS | | | | | |
| | | 1 | | 5 | | 25 | |
| Provider | Model | Client | Server | Client | Server | Client | Server |
|---|---|---|---|---|---|---|---|
| Azure | text-embedding-3-small | 0.46 | — | 0.02 | — | $1.3 \times 10^{-21}$ | — |
| Deep Infra | Llama 3.1 8B Instruct | $6.5 \times 10^{-5}$ | — | $7.5 \times 10^{-38}$ | — | | |
| Fireworks | Llama 3.1 8B Instruct | $9.0 \times 10^{-17}$ | $5.2 \times 10^{-32}$ | | | | |
| Lepton | Llama 3.1 8B Instruct | 0.12 | 0.07 | $1.2 \times 10^{-10}$ | $1.4 \times 10^{-9}$ | | |
| OpenAI | text-embedding-3-small | 0.41 | 0.36 | 0.20 | 0.08 | $1.1 \times 10^{-19}$ | $1.1 \times 10^{-19}$ |
| Perplexity | Llama 3.1 8B Instruct | $5.3 \times 10^{-74}$ | — | | | | |
| Replicate | Llama 3 8B Instruct | — | $8.6 \times 10^{-150}$ | | | | |
| Anthropic | Claude 3 Haiku | 0.24 | — | 0.77 | — | 0.87 | — |
| OpenAI | GPT-4o mini | 0.41 | 0.20 | 0.41 | 0.62 | 0.41 | 0.94 |

*Table 7.* Server-side response times and first five embedding coordinates when sending the same prompt 25 consecutive times to the OpenAI text-embedding-3-small API from the same user. Blue denotes the "normal" embedding returned in most of the responses with normal response times, which indicate cache misses. Green denotes fast response times, which indicate cache hits. Red denotes embeddings that differ from both the "normal" and "cache hit" embeddings.

| Time (s) | Embedding | | | | |
| --- | --- | --- | --- | --- | --- |
| | Coordinate 1 | Coordinate 2 | Coordinate 3 | Coordinate 4 | Coordinate 5 |
| 0.100 | 0.00522740 | 0.02509154 | $-0.04450446$ | 0.01837845 | 0.02944954 |
| 0.096 | 0.00534875 | 0.02516800 | $-0.04450600$ | 0.01841400 | 0.02952400 |
| 0.119 | 0.00534875 | 0.02516800 | $-0.04450600$ | 0.01841400 | 0.02952400 |
| 0.088 | 0.00534875 | 0.02516800 | $-0.04450600$ | 0.01841400 | 0.02952400 |
| 0.216 | 0.00523302 | 0.02509207 | $-0.04457144$ | 0.01835683 | 0.02947217 |
| 0.100 | 0.00534875 | 0.02516800 | $-0.04450600$ | 0.01841400 | 0.02952400 |
| 0.096 | 0.00534875 | 0.02516800 | $-0.04450600$ | 0.01841400 | 0.02952400 |
| 0.088 | 0.00534875 | 0.02516800 | $-0.04450600$ | 0.01841400 | 0.02952400 |
| 0.077 | 0.00534875 | 0.02516800 | $-0.04450600$ | 0.01841400 | 0.02952400 |
| 0.036 | 0.00535751 | 0.02517040 | $-0.04455426$ | 0.01839376 | 0.02954882 |
| 0.076 | 0.00534875 | 0.02516800 | $-0.04450600$ | 0.01841400 | 0.02952400 |
| 0.124 | 0.00534875 | 0.02516800 | $-0.04450600$ | 0.01841400 | 0.02952400 |
| 0.280 | 0.00522179 | 0.02509099 | $-0.04452551$ | 0.01835604 | 0.02947091 |
| 0.032 | 0.00535751 | 0.02517040 | $-0.04455426$ | 0.01839376 | 0.02954882 |
| 0.089 | 0.00534875 | 0.02516800 | $-0.04450600$ | 0.01841400 | 0.02952400 |
| 0.034 | 0.00535751 | 0.02517040 | $-0.04455426$ | 0.01839376 | 0.02954882 |
| 0.092 | 0.00534875 | 0.02516800 | $-0.04450600$ | 0.01841400 | 0.02952400 |
| 0.089 | 0.00534875 | 0.02516800 | $-0.04450600$ | 0.01841400 | 0.02952400 |
| 0.103 | 0.00534875 | 0.02516800 | $-0.04450600$ | 0.01841400 | 0.02952400 |
| 0.127 | 0.00523272 | 0.02513466 | $-0.04454690$ | 0.01837780 | 0.02944849 |
| 0.094 | 0.00534875 | 0.02516800 | $-0.04450600$ | 0.01841400 | 0.02952400 |
| 0.039 | 0.00535751 | 0.02517040 | $-0.04455426$ | 0.01839376 | 0.02954882 |
| 0.035 | 0.00535751 | 0.02517040 | $-0.04455426$ | 0.01839376 | 0.02954882 |
| 0.080 | 0.00534875 | 0.02516800 | $-0.04450600$ | 0.01841400 | 0.02952400 |
| 0.034 | 0.00535751 | 0.02517040 | $-0.04455426$ | 0.01839376 | 0.02954882 |

*Table 8.* Server-side response times and first five embedding coordinates when sending the same prompt 25 consecutive times to the OpenAI text-embedding-3-small API from the same user. Blue denotes the "normal" embedding returned in most of the responses with normal response times, which indicate cache misses. Green denotes fast response times, which indicate cache hits. Red denotes embeddings that differ from both the "normal" and "cache hit" embeddings.

| Time (s) | Embedding | | | | |
| --- | --- | --- | --- | --- | --- |
| | Coordinate 1 | Coordinate 2 | Coordinate 3 | Coordinate 4 | Coordinate 5 |
| 0.093 | 0.00455398 | 0.02148935 | −0.05159185 | 0.01970582 | 0.02810146 |
| 0.079 | 0.00455398 | 0.02148935 | −0.05159185 | 0.01970582 | 0.02810146 |
| 0.081 | 0.00455398 | 0.02148935 | −0.05159185 | 0.01970582 | 0.02810146 |
| 0.087 | 0.00455398 | 0.02148935 | −0.05159185 | 0.01970582 | 0.02810146 |
| 0.112 | 0.00455518 | 0.02146046 | −0.05157467 | 0.01972101 | 0.02807036 |
| 0.038 | 0.00453244 | 0.02149035 | −0.05159424 | 0.01968498 | 0.02810276 |
| 0.113 | 0.00455398 | 0.02148935 | −0.05159185 | 0.01970582 | 0.02810146 |
| 0.036 | 0.00453244 | 0.02149035 | −0.05159424 | 0.01968498 | 0.02810276 |
| 0.078 | 0.00455398 | 0.02148935 | −0.05159185 | 0.01970582 | 0.02810146 |
| 0.118 | 0.00455531 | 0.02148279 | −0.05153262 | 0.01974330 | 0.02809289 |
| 0.079 | 0.00455518 | 0.02146046 | −0.05157467 | 0.01972101 | 0.02807036 |
| 0.084 | 0.00455398 | 0.02148935 | −0.05159185 | 0.01970582 | 0.02810146 |
| 0.096 | 0.00455398 | 0.02148935 | −0.05159185 | 0.01970582 | 0.02810146 |
| 0.110 | 0.00455398 | 0.02148935 | −0.05159185 | 0.01970582 | 0.02810146 |
| 0.089 | 0.00455398 | 0.02148935 | −0.05159185 | 0.01970582 | 0.02810146 |
| 0.063 | 0.00453244 | 0.02149035 | −0.05159424 | 0.01968498 | 0.02810276 |
| 0.035 | 0.00453244 | 0.02149035 | −0.05159424 | 0.01968498 | 0.02810276 |
| 0.094 | 0.00455398 | 0.02148935 | −0.05159185 | 0.01970582 | 0.02810146 |
| 0.100 | 0.00455398 | 0.02148935 | −0.05159185 | 0.01970582 | 0.02810146 |
| 0.033 | 0.00453244 | 0.02149035 | −0.05159424 | 0.01968498 | 0.02810276 |
| 0.112 | 0.00455398 | 0.02148935 | −0.05159185 | 0.01970582 | 0.02810146 |
| 0.036 | 0.00453244 | 0.02149035 | −0.05159424 | 0.01968498 | 0.02810276 |
| 0.033 | 0.00453244 | 0.02149035 | −0.05159424 | 0.01968498 | 0.02810276 |
| 0.092 | 0.00455398 | 0.02148935 | −0.05159185 | 0.01970582 | 0.02810146 |
| 0.118 | 0.00454002 | 0.02149847 | −0.05158763 | 0.01983471 | 0.02807742 |

*Table 9.* Server-side response times and first five embedding coordinates when sending the same prompt 25 consecutive times to the OpenAI text-embedding-3-small API from the same user. Blue denotes the "normal" embedding returned in most of the responses with normal response times, which indicate cache misses. Green denotes fast response times, which indicate cache hits. Red denotes embeddings that differ from both the "normal" and "cache hit" embeddings.

| | Embedding | | | | |
| --- | --- | --- | --- | --- | --- |
| Time (s) | Coordinate 1 | Coordinate 2 | Coordinate 3 | Coordinate 4 | Coordinate 5 |
| 0.113 | 0.00306934 | 0.02534029 | −0.05656114 | 0.02567696 | 0.02828057 |
| 0.033 | 0.00308367 | 0.02534279 | −0.05652182 | 0.02570194 | 0.02821602 |
| 0.087 | 0.00306934 | 0.02534029 | −0.05656114 | 0.02567696 | 0.02828057 |
| 0.097 | 0.00306934 | 0.02534029 | −0.05656114 | 0.02567696 | 0.02828057 |
| 0.163 | 0.00306934 | 0.02534029 | −0.05656114 | 0.02567696 | 0.02828057 |
| 0.142 | 0.00305834 | 0.02536461 | −0.05647554 | 0.02572376 | 0.02826022 |
| 0.033 | 0.00308367 | 0.02534279 | −0.05652182 | 0.02570194 | 0.02821602 |
| 0.119 | 0.00306308 | 0.02531247 | −0.05650425 | 0.02569395 | 0.02827457 |
| 0.090 | 0.00306934 | 0.02534029 | −0.05656114 | 0.02567696 | 0.02828057 |
| 0.100 | 0.00306934 | 0.02534029 | −0.05656114 | 0.02567696 | 0.02828057 |
| 0.261 | 0.00306656 | 0.02531808 | −0.05647188 | 0.02576698 | 0.02823594 |
| 0.426 | 0.00308757 | 0.02537424 | −0.05663172 | 0.02575598 | 0.02822604 |
| 0.040 | 0.00308367 | 0.02534279 | −0.05652182 | 0.02570194 | 0.02821602 |
| 0.036 | 0.00308367 | 0.02534279 | −0.05652182 | 0.02570194 | 0.02821602 |
| 0.075 | 0.00306934 | 0.02534029 | −0.05656114 | 0.02567696 | 0.02828057 |
| 0.087 | 0.00306934 | 0.02534029 | −0.05656114 | 0.02567696 | 0.02828057 |
| 0.093 | 0.00306934 | 0.02534029 | −0.05656114 | 0.02567696 | 0.02828057 |
| 0.116 | 0.00305834 | 0.02536461 | −0.05647554 | 0.02572376 | 0.02826022 |
| 0.140 | 0.00306308 | 0.02531247 | −0.05650425 | 0.02569395 | 0.02827457 |
| 0.034 | 0.00308367 | 0.02534279 | −0.05652182 | 0.02570194 | 0.02821602 |
| 0.033 | 0.00306686 | 0.02536540 | −0.05638750 | 0.02572455 | 0.02823864 |
| 0.033 | 0.00308367 | 0.02534279 | −0.05652182 | 0.02570194 | 0.02821602 |
| 0.032 | 0.00308367 | 0.02534279 | −0.05652182 | 0.02570194 | 0.02821602 |
| 0.089 | 0.00306934 | 0.02534029 | −0.05656114 | 0.02567696 | 0.02828057 |
| 0.032 | 0.00308367 | 0.02534279 | −0.05652182 | 0.02570194 | 0.02821602 |

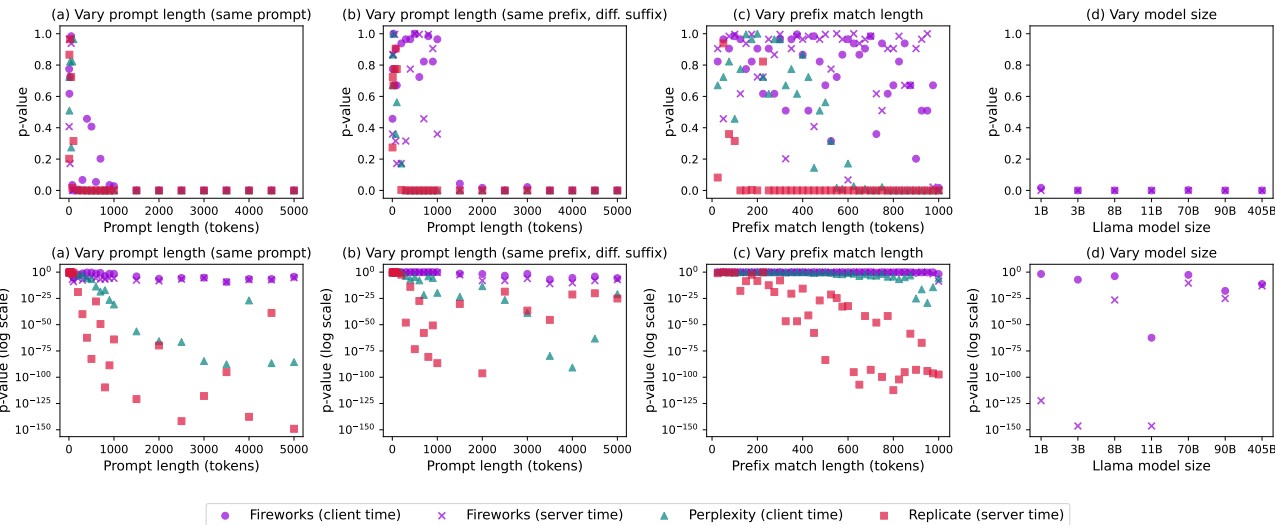

*Figure 7.* Ablations on the effects of PROMPTLENGTH, PREFIXFRACTION, and model size on the audit p-values. Each test is run using NUMSAMPLES = 250. The top and bottom rows display the p-values on linear and logarithmic scales, respectively. In (a)–(c), as the prompt length or prefix match length decreases, the p-values grow larger. In (d), we detect caching across all model sizes, with no clear relationship between model size and p-values.

