# OpenReview forum: "Auditing Prompt Caching in Language Model APIs"
_ICML.cc/2025/Conference — ICML 2025 poster_

### Official Review · Reviewer_iqr7 · 2025-03-06

**Overall Recommendation:** 3

**Summary:**

This paper ascertains that hint caching in the LLM API can precipitate privacy breaches and divulge model architecture information. To prevent such issues, API providers are recommended to permit only user-level caching and make caching policies public for enhanced transparency. Through these examinations, the authors aim to continue evaluating and auditing the security and privacy of machine learning systems. Using empirical research, the paper reveals the risks of prompt caching and proposes corresponding mitigation strategies, providing a valuable reference for related work.

**Claims And Evidence:**

The comparative experiments are relatively comprehensive.

**Essential References Not Discussed:**

Not yet.

**Ethical Review Concerns:**

Not yet.

**Experimental Designs Or Analyses:**

The paper conducts experiments on prompt caching in language model APIs. It targets 17 API providers, constructs cache - hit and cache - miss procedures with different parameters, uses statistical hypothesis testing and two - sample KS test to determine caching status, cache - sharing levels, evaluate attackers' discrimination ability, and explore information leakage.

**Methods And Evaluation Criteria:**

The method proposed in this paper can be used to detect the existence of prompt cache and the level of cache sharing. This approach is useful for understanding the behavior of LLMs in real-world applications, especially in privacy-focused scenarios, where the audits mentioned in the paper are conducted over multiple actual API calls, suggesting that the approach is closely related to real-world application scenarios.

**Other Comments Or Suggestions:**

Not yet.

**Other Strengths And Weaknesses:**

**Strengths**：
1. The paper centers on the significant but overlooked prompt caching in language model APIs, exploring privacy and architecture info leakage.
2. Its well - designed cache - hit/miss experiments with statistical testing can precisely identify API caching and sharing levels, offering a practical security - detection tool.
3. The experiments involve diverse models, representing different APIs' prompt - caching realities.

**Weaknesses**：
1. Complex request - routing strategies may distort experimental results and mislead caching judgments.
2. The Bonferroni correction in multiple testing is too conservative, potentially missing real caching and underestimating security risks.
3. The paper fails to deeply verify other factors that could cause similar caching phenomena.

**Questions For Authors:**

Not yet.

**Relation To Broader Scientific Literature:**

The key contribution of this paper is the development of an audit method to detect the phenomenon of prompt caching in language model apis and identify different levels of prompt sharing patterns, including global, organizational, and individual levels, providing a new perspective on the design and implementation of language model APIs.

**Theoretical Claims:**

The article basically does not involve theoretical proof.

---

> ### Author Rebuttal · Authors · 2025-04-01
>
> Thank you for the detailed and thoughtful review! We address the main points below.
>
> ## 1. Request routing strategies
>
> We can show that our experimental results are valid regardless of routing strategies. Under the null hypothesis $H_0$ of no caching, request-routing will be independent of whether the prompt was recently sent. Therefore, since the distribution of prompts $\mathcal{P}$ is the same in the cache hit and cache miss procedures (these procedures differ only in whether the attacker’s prompt was previously sent), a non-caching request-routing strategy will perform the same on the distributions of the cache hit and cache miss procedures. Therefore, $\mathcal{D}\_\text{hit} = \mathcal{D}\_\text{miss}$ still holds under the null hypothesis $H_0$, so the statistical audit is valid.
>
> If the API provider is performing caching, then different routing strategies may make it easier or harder to detect caching, but the audit still outputs valid p-values with respect to the null hypothesis. For example, if prompts are intentionally routed to a server where the prompt is already cached, it will be easy to produce and detect cache hits using `NumVictimRequests = 1`. On the other hand, if prompts are randomly routed, more victim requests may be needed to detect caching. Multiple victim requests may be needed to cache the prompt in enough servers for the attacker’s prompt to have a sufficient probability of producing a cache hit.
>
> ## 2. Bonferroni correction
>
> While the Bonferroni correction is indeed conservative, it is simple and straightforward, and we only use a maximum correction factor of 6. We are confident that performing Bonferroni correction did not cause our audits to miss any real caching because the p-values for the APIs in which we did not detect caching were orders of magnitude larger than the significance level of $\alpha = 10^{-8}$. As shown in Table 3 in the appendix (page 14), in the first level of audits, all non-significant p-values were larger than 0.1, and most significant p-values were many orders of magnitude smaller than $10^{-8}$.
>
> ## 3. Other factors that could cause similar caching phenomena
>
> For the purposes of our audit, the specific caching mechanism is unimportant, as long as it follows the simple properties we describe (page 2, line 70, right column). The potential privacy leakage does not depend on what specific mechanisms cause timing differences between cached and non-cached prompts; the leakage occurs simply because there *are* timing differences. Even if the caching phenomena has strange causes—e.g., the server intentionally delays responses for new prompts but immediately returns responses for previously seen prompts—our audits can detect it, and the timing differences lead to potential privacy leakage.

---

> > ### Comment · Reviewer_iqr7 · 2025-04-05
> >
> > Thank you for your efforts in rebuttal. Parts of my concerns are addressed.
> >
> > However, the practical applicability of the proposed leakage seems difficult to achieve.
> >
> >  I decide to maintain my score.

---

### Official Review · Reviewer_KxRJ · 2025-03-11

**Overall Recommendation:** 3

**Summary:**

This paper investigates the privacy leakage caused by the prompt caching in LLMs. Basically, prompt caching improves the efficiency of inference by caching and reusing the internal results of previous prompts. The attack could infer whether a given prompt has been used (cached) by simply checking the time to the first token (TTFT).
The paper provides an extentive evaluation to support its claim. It also proposes some mitigation strategies which are reasonable and insightful.

**Claims And Evidence:**

Most claims made in this paper are reasonable and supported by existing works or evaluation results.

**Essential References Not Discussed:**

None.

**Experimental Designs Or Analyses:**

The experimental designs and analyses make sense to me.

**Methods And Evaluation Criteria:**

Yes.

**Other Comments Or Suggestions:**

This paper explores a novel privacy leakage lead by the prompt caching techniques in most recent LLMs.
The claims are mostly correct and reasonable, supported by a series of evaluations.

I appreciate the disclose of the potential issues and the discussion of countermeasures.
My primary concern is that the practical exploitability of the proposed leakage is not easy, as it's hard to guess out the prompt prefix with a nontrival percentage of the entire length. Therefore, the leakage may not be that significant if it is not easily exploited. I suggest the authors could pay some attention on this part.

**Other Strengths And Weaknesses:**

Strengths:

- The code and experimental data are provided.
- The writing is good and easy to follow.

Weakness:

- Limited discussion of practical exploitability of the proposed leakage.

**Questions For Authors:**

Please see above.

**Relation To Broader Scientific Literature:**

This paper provides a thorough analysis of privacy and security vulnerabilities associated with prompt caching in LLM APIs. It effectively highlights potential risks and discusses mitigation strategies. The findings have significant implications, raising awareness about security concerns in model deployment and emphasizing the need for preventive measures.

**Theoretical Claims:**

There is no proof in this paper. It is mostly an empirical paper based on existing statistical methods.

---

> ### Author Rebuttal · Authors · 2025-03-31
>
> Thank you for the detailed and thoughtful review! We address the main points below.
>
> ## Practical exploitability
>
> We agree that practical exploitations of prompt cache sharing are challenging. The attacker needs to guess a long prompt prefix to check if it is cached, as you described. As we discussed in Section 4.4: Difficulty of Prompt Extraction Attacks (page 7, line 350, left column), a natural idea is to use breadth-first search to try to extract cached prompts token-by-token. However, we were unable to execute practical prompt extraction attacks due to several challenges, such as the difficulty of making repeated measurements. Accordingly, we emphasized that the privacy leakage is only potential throughout the paper.
>
> However, we believe that even potential privacy leakage due to global cache sharing is a cause for concern, especially as LLM APIs are being used by increasingly many users and companies for increasingly many tasks, which may include sensitive data. In addition, future work may overcome or eliminate the practical challenges we discussed.
>
> Following our responsible disclosure, several API providers made changes to mitigate the potential privacy leakage. OpenAI, Microsoft Azure, and Fireworks AI worked quickly to mitigate vulnerabilities by stopping global cache sharing. (Note that this is not an exhaustive list of all companies that implemented fixes.) Fireworks AI also added detailed [documentation](https://docs.fireworks.ai/guides/prompt-caching) about prompt caching and data privacy, as well as an option to opt-out of caching for a particular prompt. The mitigations implemented by these companies illustrate the real-world impact of our findings.
>
> Also, we found that prompt caching can leak information about model architecture, which is of practical importance given the competitiveness and secrecy of the modern LLM landscape. Namely, we found evidence that OpenAI’s text-embedding-3-small model has a decoder-only Transformer architecture, which was previously not publicly known.

---

> > ### Comment · Reviewer_KxRJ · 2025-04-06
> >
> > I appreciate the authors' response.
> > While I believe the finding highlights a potential privacy leakage, I still find it somewhat challenging to exploit in practice.
> >
> > I acknowledge the value of the authors’ insights and contributions, and I will maintain my score as a weak accept.

---

### Official Review · Reviewer_uAtv · 2025-03-14

**Overall Recommendation:** 4

**Summary:**

The paper explores a novel privacy risk for hosted language models: timing attacks on the host's prefix cache. They use hypothesis testing to demonstrate that popular LM providers are indeed using prefix caching.

**Claims And Evidence:**

I believe all of the claims made are well supported with statistical evidence and findings that are not completely certain are appropriately presented.

**Essential References Not Discussed:**

I am not aware of any essential works that were not cited properly in the paper.

**Experimental Designs Or Analyses:**

The experimental design for public LM api providers is very sound. One potentially interesting result that is not included is testing in a self-hosted API server which can be configured to support prefix caching (like the one provided by VLLM). This may help demonstrate how easy it is to detect prefix caching in an ideal setting.

**Methods And Evaluation Criteria:**

The method proposed looks like exactly the right thing to do and the evaluation of the test on popular LM providers is of great significance to the community.

**Other Comments Or Suggestions:**

The authors responsible disclosure of the detected vulnerability to the model providers is commendable.

**Other Strengths And Weaknesses:**

Other strength: The framing of audit in terms of hypothesis testing is very natural!

**Questions For Authors:**

Another technique that may affect the TTFT is speculative decoding. Do you think there is any potential to try to infer the speculation behavior in order to help "denoise" the timing results and focus on the impact of the cache?

Do you think LM providers could strategically delay their request responses so the timing is identical between cached and uncached requests, while still saving on compute cost?

**Relation To Broader Scientific Literature:**

This paper is the latest in a line of work on attacking models via their APIs, many of which are detailed in the related work section. The specific attack of this paper has not been addressed by any prior work. It represents a significant new attack which will hopefully influence model providers to carefully design their caching systems with properly enforced boundaries.

**Theoretical Claims:**

There are no major theoretical claims made.

---

> ### Author Rebuttal · Authors · 2025-03-31
>
> Thank you for the detailed and thoughtful review! We address the questions below.
>
> ## Speculative decoding
>
> This is an interesting point. We have thought about this, and we believe that speculative decoding should not impact the TTFT when we set the max response tokens to 1. First, we note that speculative decoding is not beneficial when only 1 response token is generated. Speculative decoding is only beneficial when the smaller draft model can generate multiple tokens, and the larger target model can “check” these tokens in parallel. Since the larger target model has to make a forward pass regardless, when generating only 1 token, it is faster to skip the smaller draft model.
>
> If speculative decoding is nevertheless enabled when generating only 1 token, we believe that it would not cause a noticeable variation in TTFT across different prompts (of the same length). In this scenario, the smaller draft model would generate 1 token, then the larger target model would make a forward pass and either accept or reject the draft token. The only timing difference between accepting and rejecting would come from resampling a token from the target distribution, which is negligible compared to the time for the forward pass.
>
> Note that when the LLM generates full responses (number of tokens $\gg 1$), speculative decoding causes data-dependent timing variations that may be exploited. As mentioned in the related works (page 8, line 404, right column), Carlini & Nasr (2024) and Wei et al. (2024) exploit speculative decoding to extract encrypted and streamed LLM responses by measuring delays between packets.
>
> ## Intentionally delaying responses
>
> Yes, we believe that intentionally delaying the response times for cache hits so that they look like cache misses is a viable mitigation for providers. We briefly touch upon this in the paper (page 8, line 410, left column). This eliminates the benefits of prompt caching for users, but API providers could still benefit, as cached prompts require less GPU processing time.
>
> Providers would need to be somewhat careful about implementing this, as simply waiting a random amount of time may not adequately disguise cache hits. One better strategy is to first compute distributions of the server-side TTFT for cache misses for various prompt lengths. Then, when a cache hit occurs, the server would sample a TTFT from the cache miss distribution corresponding to the given prompt length, and delay the response until that TTFT has elapsed (if the actual TTFT has already exceeded the sampled TTFT, then send the response immediately). This way, the distribution of times for cache hits and cache misses should approximately match each other.

---

> > ### Comment · Reviewer_uAtv · 2025-04-05
> >
> > I would like to thank the authors for their rebuttal. I maintain that the paper should be accepted, so I will keep my score.

---

### Official Review · Reviewer_6FQk · 2025-03-20

**Overall Recommendation:** 3

**Summary:**

This paper presents an empirical audit of prompt caching mechanisms in language model APIs. It demonstrates that timing differences, arising from cache hits and cache misses, can potentially leak private information and even reveal details about a model’s architecture. The study employs statistical hypothesis testing across various real-world API providers to characterize different levels of cache sharing (per-user, per-organization, and global). The results indicate significant vulnerabilities, including potential privacy leakage and exposure of proprietary model details.

**Claims And Evidence:**

The paper makes strong claims regarding the vulnerability of LLM APIs to timing side-channel attacks via prompt caching. It supports these claims with rigorous statistical evidence. However, some evidence is derived under assumptions, such as using random prompts to simulate cache misses, that may not universally hold in real-world deployments. This reliance on assumptions could limit the generalizability of the findings, and further empirical validation is needed.

**Essential References Not Discussed:**

The mentioned related works are essential to understanding the main contributions of this paper.

**Ethical Review Concerns:**

The paper highlights significant privacy vulnerabilities and the potential for misuse of timing side-channel attacks, which may allow attackers to infer sensitive information from user prompts.

**Ethical Review Flag:**

Flag this paper for an ethics review.

**Ethics Expertise Needed:**

["Responsible Research Practice (e.g., IRB, documentation, research ethics, participant consent)"]

**Experimental Designs Or Analyses:**

The experimental design is methodically sound, with clear separation between the cache hit and miss procedures and extensive use of statistical tests. However, the reliance on a synthetic prompt distribution (i.e., random sequences of tokens) to simulate cache misses is a weakness. It remains unclear whether these conditions reflect typical user inputs. The experiments would be more convincing if supplemented with tests involving natural language prompts or real-world API logs, which could validate the assumptions underlying the timing measurements.

**Methods And Evaluation Criteria:**

The methodology, including the construction of cache hit and miss procedures and the application of nonparametric statistical tests, is innovative and well-structured for the research problem. Nevertheless, the evaluation relies on assumptions (e.g., that random prompts always produce cache misses) that are difficult to verify. Additional experiments using more realistic or varied prompt distributions, along with evaluation criteria that mimic real-world conditions, would strengthen the paper’s claims.

**Other Comments Or Suggestions:**

I recommend including an explicit ethical statement that addresses the potential misuse of the research findings. Additionally, integrating a prototype or a detailed case study that illustrates a practical exploitation scenario would greatly enhance the paper’s impact. Finally, further empirical validation of the assumptions regarding prompt distribution and cache behavior is necessary to reinforce the generality of the conclusions.

**Other Strengths And Weaknesses:**

Strengths:

- The paper tackles an important and timely security issue in AI.
- The statistical framework is rigorous, and the empirical analysis is detailed.
- The categorization of cache sharing levels is insightful and adds nuance to the discussion.

Weaknesses:

- Heavy reliance on assumptions regarding prompt distributions (e.g., that random prompts always yield cache misses.
- The experimental setup lacks naturalistic workloads that reflect how APIs are used in practice.
- Some discussion could be more in-depth, particularly with regard to potential exploitation scenarios and mitigation strategies.

**Questions For Authors:**

- Can you provide empirical evidence that the assumption of random prompts reliably produces cache misses holds true in real-world API usage scenarios?

- Have you considered evaluating your attack model using natural language prompts or actual API traffic to better mimic realistic conditions?

- Would the authors be able to include a proof-of-concept demonstration or a detailed case study that illustrates a potential real-world exploitation of these vulnerabilities?

**Relation To Broader Scientific Literature:**

The paper positions itself well within the landscape of cache timing attacks and side-channel vulnerabilities. It is also motivated by recent developments in LLM optimization and inference acceleration.

**Theoretical Claims:**

While the paper does not focus heavily on formal proofs, it builds a insightful theoretical basis for linking cache timing differences to potential privacy leaks.

---

> ### Author Rebuttal · Authors · 2025-03-31
>
> Thank you for the detailed and thoughtful review! We address the main points below.
>
> ## Random prompts produce cache misses
>
> We assume that random prompts produce cache misses because it is exceedingly unlikely that a random prompt shares a prefix of noticeable length with any cached prompts. In the worst case, assume that all other users are sending random prompts with the same structure as in the paper, i.e., random letters separated by spaces. (In reality, very few, if any, other users will be sending such prompts.) As mentioned in the paper (page 4, line 196, left column), the probability that two of these random prompts share a prefix of 15 tokens or longer is less than $10^{-25}$. Assume that the server can store 1 billion prompts in its cache (in reality, the true cache capacity is likely much smaller). Then, using a union bound, the probability that a random prompt shares a prefix of 15 tokens or longer with any cached prompt is less than $10^{-25} \times 10^{9} = 10^{-16}$. We send 250 random prompts for cache miss timings in each audit, so using another union bound, the probability that any of these prompts produce cache hits is less than $10^{-12}$.
>
> We are also able to empirically confirm that random prompts produce cache misses. Some API providers have officially released prompt caching features, such as [OpenAI](https://platform.openai.com/docs/guides/prompt-caching) and [Anthropic](https://docs.anthropic.com/en/docs/build-with-claude/prompt-caching). As part of these features, each API response returns the number of cached prompt tokens. Using this information, we ran some simple tests that empirically confirm that the random prompts used in our experiments consistently produce cache misses in real-world API usage scenarios.
>
> ## Random prompts versus natural language
>
> We used random alphabetic prompts instead of natural language prompts for a few reasons. We can assume that random prompts reliably produce cache misses, as discussed above. However, there is a greater chance that prefixes of realistic natural language prompts have already been sent by other users, making it harder to reliably measure cache misses. In addition, it is more difficult to construct a clean, well-defined distribution of natural language prompts containing exactly `PromptLength` tokens, compared to using random prompts. This distribution is important for our rigorous statistical hypothesis testing framework.
>
> As mentioned in the paper (page 6, line 302, right column), our audits detected the exact level of cache sharing stated by OpenAI and Anthropic for their chat models (per-organization sharing, but not global sharing). This demonstrates the efficacy of our audits on real-world APIs, even if the random prompts do not necessarily reflect realistic workflows.
>
> ## Real-world exploitation and mitigations
>
> As discussed in Section 4.4: Difficulty of Prompt Extraction Attacks (line 350, page 7, left column), we believe that real-world exploitation of these vulnerabilities is challenging. However, we believe that even potential privacy leakage due to global cache sharing is a cause for concern, especially as LLM APIs are being used by increasingly many users and companies for increasingly many tasks, which may include sensitive data. In addition, future work may overcome or eliminate the practical challenges we discussed.
>
> To mitigate these vulnerabilities, we believe that API providers should disable global cache sharing and disclose the level of cache sharing. Following our responsible disclosure, several API providers followed this approach. OpenAI, Microsoft Azure, and Fireworks AI worked quickly to mitigate vulnerabilities by stopping global cache sharing. (Note that this is not an exhaustive list of all companies that implemented fixes.) Fireworks AI also added detailed [documentation](https://docs.fireworks.ai/guides/prompt-caching) about prompt caching and data privacy, as well as an option to opt-out of caching for a particular prompt. The mitigations implemented by these companies illustrate the real-world impact of our findings.
>
> Also, we found that prompt caching can leak information about model architecture, which is of practical importance given the competitiveness and secrecy of the modern LLM landscape. Namely, we found evidence that OpenAI’s text-embedding-3-small model has a decoder-only Transformer architecture, which was previously not publicly known.
>
> ## Ethics review
>
> As discussed in the Impact Statement (page 9, line 440, left column), to mitigate real-world harms arising from our research, we followed standard responsible disclosure practices for security vulnerabilities. In October 2024, we disclosed our audit results with each API provider in which we detected prompt caching. We gave providers 60 days to address the vulnerabilities before publicly releasing or submitting our findings, and the actual time elapsed ended up being longer.

---

### Decision · Program_Chairs · 2025-05-01

**Decision:**

Accept (poster)

**Comment:**

This paper introduces a timely and impactful empirical audit of prompt caching behavior in commercial LLM APIs, revealing potential privacy and security vulnerabilities through carefully designed timing-based side-channel analysis. The methodology is rigorous and well-motivated, grounded in statistical hypothesis testing, and the evaluation is thorough across a broad range of providers. Despite relying on assumptions such as random prompts generating cache misses, the authors provide both theoretical justifications and empirical confirmation that lend credibility to their setup. The work has already prompted real-world impact through responsible disclosure, with several providers implementing mitigations in response. While the practical exploitability of the described leakage may be limited today, the paper makes a compelling case that even potential leakage is significant as LLMs become more widely used in sensitive settings. Overall, despite some limitations in assumptions and generalizability, the contributions are novel, the findings are important, and the paper should be included in the conference.